# The association between depressive symptoms and antibody response following SARS-CoV-2 vaccination among central North Carolina residents

Caitlin A. Cassidy[1]*, Jessie K. Edwards[1,2], Annika K. Gunderson[1,2], Haley Abernathy[1], Haley E. Garrett[3], Elise King[4], Cherese N. Beatty Parker[5,6], Anne P. Starling[1], Emily J. Ciccone[4], Ross M. Boyce[1,2,4], Bonnie E. Shook-Sa[7], Allison E. Aiello[5,6]

1 Department of Epidemiology, Gillings School of Global Public Health, The University of North Carolina at Chapel Hill, Chapel Hill, North Carolina, United States of America, 2 Carolina Population Center, The University of North Carolina at Chapel Hill, Chapel Hill, North Carolina, United States of America, 3 Department of Genetics, School of Medicine, The University of North Carolina at Chapel Hill, Chapel Hill, North Carolina, United States of America, 4 Division of Infectious Diseases, School of Medicine, The University of North Carolina at Chapel Hill, Chapel Hill, North Carolina, United States of America, 5 Department of Epidemiology, Mailman School of Public Health, Columbia University, New York, New York, United States of America, 6 Robert N Butler Columbia Aging Center, Mailman School of Public Health, Columbia University, New York, New York, United States of America, 7 Department of Biostatistics, Gillings School of Global Public Health, The University of North Carolina at Chapel Hill, Chapel Hill, North Carolina, United States of America

* cacassid@email.unc.edu

## Abstract

A large body of research suggests that stress can affect how the immune system responds to vaccines. The impact of mental health disorders on humoral antibody response following immunization is not well understood, particularly for the COVID-19 vaccines. Leveraging a population-based longitudinal cohort assessing SARS-CoV-2 incidence in central North Carolina, we sought to investigate the relationship between mental health and immune response to vaccination. The 106 participants responded to biweekly surveys and contributed monthly serum samples that underwent SARS-CoV-2 spike antibody testing via enzyme-linked immunosorbent assay (ELISA). Utilizing weighted generalized linear models, we measured the association between depressive symptoms as recorded on the CESD-20 and quantitative antibody levels after COVID-19. Overall, we found modest differences in post-vaccination IgG between participants with depression and those without. Individuals with depressive symptoms had smaller initial antibody responses after vaccination (mean difference = -3.55, 95% CI = (-12.01, 4.90)). These results suggest that mental health disorders may affect immune response to vaccination.

**Data availability statement:** Deidentified individual data that supports the results will be shared, provided the investigator who proposes to use the data has approval from an Institutional Review Board (IRB), Independent Ethics Committee (IEC), or Research Ethics Board (REB), as applicable, and executes a data use/sharing agreement with UNC. Principal Investigator should submit a request to the UNC Industry Contracting team (OSPContracting@unc.edu) to initiate the data use/sharing agreement.

**Funding:** This work was supported by an award from the North Carolina Department of Health and Human Services (NC DHHS) to R.M.B and the National Institutes of Health (P30AI050410 to B.E.S., R01AI157758 to J.E.K., and T32AI070114 to C.A.C.). The content is solely the responsibility of the authors and does not necessarily represent the official views of the NCDHHS or the National Institutes of Health. The funders had no role in study design, data collection and analysis, decision to publish, or preparation of the manuscript.

**Competing interests:** The authors have declared that no competing interests exist.

## 1. Introduction

On March 11, 2020, the World Health Organization declared COVID-19, the disease caused by the virus SARS-CoV-2, a pandemic [1]. The development of several COVID-19 vaccines has offered older-age recipients protection against severe disease and death [2–6]. Nonetheless, the rapid waning of antibody response after vaccination has been well documented [7–10]. The level of immunoglobulin G (IgG) antibodies in the body tend to decrease after vaccination over a period of several months with many individuals seroconverting from positive to negative within six months [7,11].

There is evidence that poor mental health prior to vaccination, including issues such as stress, depression, and loneliness, can impair the immune response to a vaccine [12–14]. These studies suggest that poor mental health exacerbates the waning immunity after vaccination. Psychosocial factors including sleep duration and social cohesion have been shown to be associated with antibody response after SARS-CoV-2 vaccination [15,16]. Additionally, some research suggests that psychological interventions can improve vaccine efficacy, highlighting the role that mental health can play in antibody response following vaccination [12]. In studies of other vaccines, depression and stress have been associated with lower antibody responses and higher odds of loss of antibody protection [17–19]. Yet, other studies have found no difference in immune response to vaccination between individuals with and without depression and stress [19].

It is hypothesized that psychosocial stress that occurs even several days or weeks after vaccination can shape antibody response [18]. The COVID-19 pandemic exacerbated mental health disorders, including depression, in populations around the world, particularly in the earlier years of the pandemic [20,21]. While poor mental health has already been identified as a risk factor for severe COVID-19 disease and mortality [22,23], very few studies have evaluated the association between poor mental health and antibody response after COVID-19 vaccination.

We sought to determine whether there is an association between depressive symptoms and the initial antibody response to vaccination or an association between depressive symptoms and antibody waning following vaccination. To accomplish this, we leveraged a unique longitudinal cohort study conducted in central North Carolina (NC), known as the Chatham County COVID-19 Cohort (C4). This study was initiated to estimate the prevalence and incidence of SARS-CoV-2 infection in Chatham County, but the study also provides an excellent opportunity to study antibody response in this population.

## 2. Methods

### 2.1. Ethics statement

Study protocols and procedures were approved by the University of North Carolina at Chapel Hill Institutional Review Board (#20–1632). Written informed consent was obtained from all participants.

## 2.2. Study design

In response to the pandemic, many seroprevalence studies were initiated to monitor community transmission, including several in NC [24–26]. The C4 study was a longitudinal, prospective, population-based study, the details of which have been previously published [27,28]. In brief, participants were enrolled via a stratified, two-stage cluster design. Phase 1 participants were recruited from the existing and ongoing Chatham County Community Cohort, and Phase 2 participants were recruited from additional census blocks sampled under the same methodology as the original cohort. Phase 2 households were selected for the C4 study using address-based sampling methods [29]. Participants were enrolled between August 2020 and June 2022. Participants provided monthly serum samples and completed surveys during follow-up. Serum samples were obtained via venous phlebotomy by trained phlebotomists, or at home using a Tasso serum self-collection device, which is an accurate alternative method of serum collection [30]. Serum samples were tested using enzyme-linked immunosorbent assay (ELISA) to detect SARS-CoV-2 spike antibodies in plasma, including the IgG positive/negative (P/N) ratio. The full details of the study design and protocol have been published previously [28].

## 2.3. Inclusion criteria

Participants who were enrolled between August 2020 and December 2021 were screened for inclusion in the analysis. Participants included in the analysis must have been selected via the probability design, been fully vaccinated and reported vaccination details. They also must have completed at least one survey with the depressive symptoms questionnaire within four weeks prior to their first vaccine dose and have provided at least one serum sample following their final vaccine dose to be included in the analysis.

## 2.4. Definitions

Demographic variables, including age and biological sex, were collected via a baseline survey (S1 File) administered to participants after enrollment. Age was calculated as the difference between the participant's date of birth and the date of completion of baseline survey. Health behaviors were self-reported on the baseline survey. Current smoking status was a binary variable denoting whether a participant smokes cigarettes, cigars, or a pipe daily. Current alcohol use was a binary variable denoting whether a participant currently drinks alcohol at least once a week. Exercise was a binary variable denoting whether a participant engages in physical activity, defined as an activity that raises the heart rate, at least once per week. After the rollout of vaccines, baseline and biweekly surveys were updated to include questions about vaccination status. Vaccine type as well as date of doses received were self-reported.

In biweekly surveys (S2 File), participants were asked about any depressive symptoms they were experiencing in a 20-question survey, which was scored from zero to 60 points according to the Center for Epidemiologic Studies Depression Scale (CESD-20) [31]. For each completed survey, a CESD-20 score was calculated and denoted as binary based on the recommended cutoff of 16 points [32]. This cutoff is frequently chosen to maintain relatively high sensitivity and specificity and is appropriate in most populations [32]. A score of 16 or greater was considered positive for depressive symptoms and scores between zero and 15 were considered negative for depressive symptoms.

The post-vaccination period was defined as the time following more than 14 days after the receipt of the second dose of an mRNA vaccine, Moderna (mRNA-1273) or Pfizer (BNT162b2), or more than 30 days after the receipt of the single dose Janssen (Ad26.COV2.S) vaccine to allow a full immune response to develop. The month post-vaccination was defined by 30-day increments following the receipt of the second dose of an mRNA vaccine or the receipt of the single dose Janssen vaccine (S1 Fig). Month zero refers to the period between 15- and 30-days post-vaccination for the mRNA vaccines, and antibody levels at this time are considered to represent the initial antibody response to vaccination.

## 2.5. Statistical analyses

In the primary analysis, IgG antibody levels measured for up to six months post-vaccination were modeled using weighted generalized estimating equations (GEE) to estimate the parameters of a generalized linear model. Samples collected more than six months after receipt of the final vaccine dose were removed from the analysis due to potential effects of booster doses. The outcome in the model was IgG at each month post vaccination. The exposure in the model was the binary pre-vaccination CESD-20 score. This depressive symptoms score was calculated as the average CESD-20 score from all surveys completed within 4 weeks prior to vaccination. All models were adjusted for confounding variables that were identified using a directed acyclic graph. These included: biological sex, age group (18–49 vs 50+), current smoking status (yes or no), current alcohol use (yes or no), and whether the participant exercises regularly (yes or no). A term for the interaction between months post-vaccination and depressive symptoms was included to allow antibody levels to vary between those with and without depressive symptoms at each month. An exchangeable working correlation was assumed to allow for repeated measurements in the same participant to be equally correlated with each other. Sampling weights derived during study sampling procedures were calibrated using general exponential models to account for nonresponse [33]. The weights were calibrated to 2020 Census population estimates in Chatham County, stratified by age group (18–49 vs 50+) and biological sex.

Missing outcomes (IgG values) and covariates were non-monotone and assumed to be missing at random (MAR). These were imputed via multiple imputation with fully conditional specification using 30 burn-in iterations and 30 imputations. The final imputation model included all confounding variables. The dates of collection for the imputed values were approximated using 30-day intervals from the participant's last true collection date. Mean IgG and corresponding 95% confidence intervals were calculated at months 0, 3, and 6, adjusted for all variables in the model. In a secondary analysis, a model was constructed as above but instead using continuous CESD-20 score as the exposure, which was calculated as the average CESD-20 score from all surveys completed within 4 weeks prior to vaccination. All analyses were conducted using SAS Studio 3.8.

## 3. Results

### 3.1. Study population

Between August 2020 and December 2021, 165 participants were recruited and enrolled into the C4 study [28]. A large percentage (80.9%) of the study population completed the initial COVID-19 vaccine series. Of these, 106 participants met the inclusion criteria to be included in these analyses (Fig 1). The most common reasons for exclusion were not providing vaccination status or being unvaccinated, not completing the CESD-20 within 4 weeks prior to the first vaccine dose, and not providing at least one serum sample after vaccination. Briefly, most participants were female (62.3%), White (87.0%), and non-Hispanic (89.2%) (Table 1). Most study participants were older than 50 years of age (75.5%) with a median age of 62 years (IQR: 50, 69). Most participants received a college or graduate degree (80.2%), and roughly half of participants reported an income of at least $75,000 per year (51.5%). A small percentage of participants reported current or previous cancer diagnoses (20.4%), diabetes mellitus (11.3%), or immunosuppression (14.3%). Of note, approximately 71% of participants resided in census defined rural areas, which aligns with the geographic distribution of Chatham County, NC (65.9% rural in 2010 Census) [34]. For initial vaccination, 102 participants received a two-dose mRNA vaccine, with 53.8% receiving Moderna and 42.5% receiving Pfizer. Four participants (3.8%) received the one-dose Janssen vaccine. More participants provided a serum sample in in the second month post-vaccination than any other month (96.2%).

### 3.2. Depressive symptoms and antibody response

Over the course of the study, many participants stopped returning for clinic visits, and thus there were fewer serum samples collected at month six than in the first 30-day period following vaccination (month zero). Prior to vaccination, 14

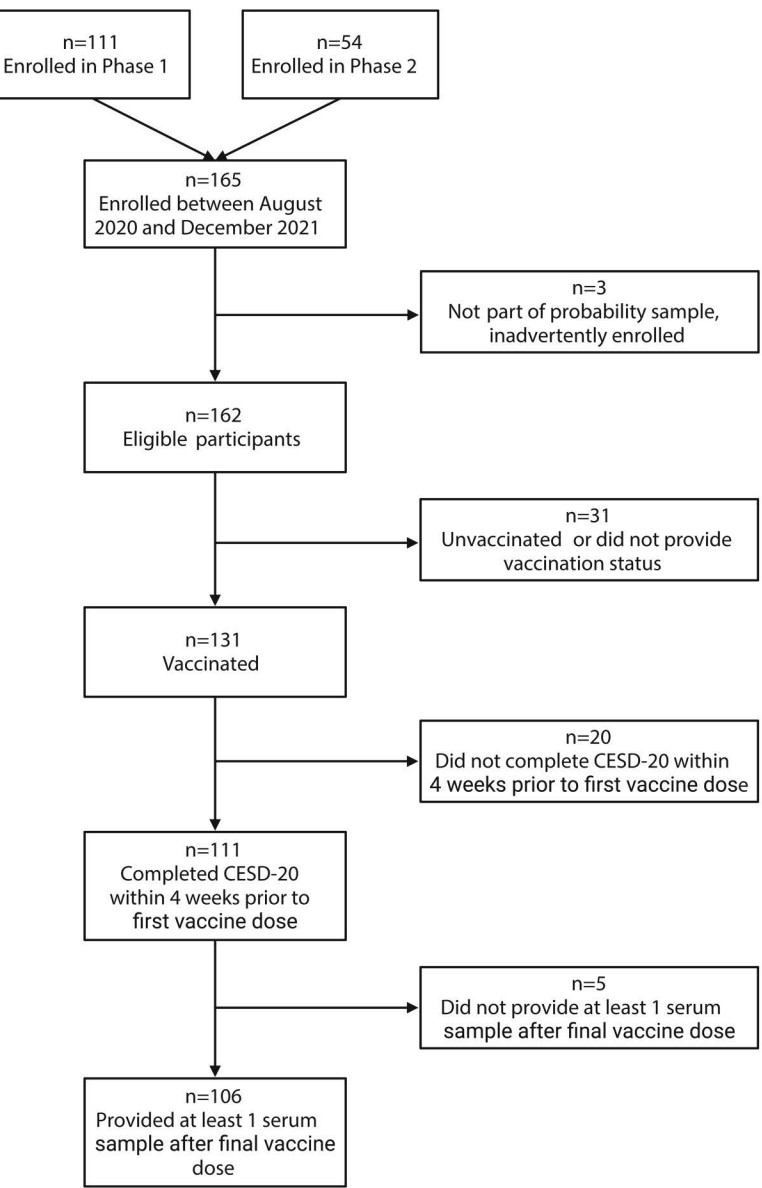

**Fig 1. Flow chart outlining inclusion criteria.**

participants (13.2%) had a positive binary CESD-20 score (i.e., ≥16) and 92 (86.8%) had a negative score. In month zero, participants without depressive symptoms had higher mean IgG levels (mean = 19.20, 95% CI=(12.33, 26.07)) than participants with depressive symptoms (mean = 15.65, 95% CI=(6.49, 24.80)). By month six, the mean IgG levels between the two groups were more similar, with the group with depressive symptoms having slightly larger mean IgG than the group without depressive symptoms (mean difference = 0.32, 95% CI=(-8.90, 9.55)). Note, the 95% confidence intervals for the differences at months 0, 3, and 6 include the null (Table 2, Fig 2). The waning of antibodies over time was observed in both groups, however the decrease was relatively small over the six-month period. Model parameter estimates are available in S1 Table. In the secondary analysis, pre-vaccination continuous CESD-20 score was negatively associated with mean IgG (parameter estimate = -0.243, 95% CI=(-0.485, -0.001)).

**Table 1. Demographic characteristics of 106 study participants.**

| Characteristic | n (%) |
|---|---|
| Age group | |
| <50 | 26 (24.5) |
| 50+ | 80 (75.5) |
| Biological sex | |
| Female | 66 (62.3) |
| Male | 40 (37.7) |
| Race | |
| White | 87 (87.0) |
| Black or African American | 6 (6.0) |
| Other | 7 (7.0) |
| Missing | 6 |
| Ethnicity | |
| Hispanic or Latino | 4 (3.9) |
| Not Hispanic or Latino | 91 (89.2) |
| Other | 7 (6.9) |
| Missing | 4 |
| Comorbidities (current or previous) | |
| Diabetes mellitus | 11 (11.3) |
| Missing | 9 |
| Cancer | 20 (20.4) |
| Missing | 8 |
| Weakened immune system (e.g., HIV, organ transplant) | 14 (14.3) |
| Missing | 8 |
| Education | |
| Less than or some college | 21 (19.8) |
| College degree or higher | 85 (80.2) |
| Income | |
| <$50,000 | 21 (20.8) |
| $50,000-$74,999 | 28 (27.7) |
| $75,000+ | 52 (51.5) |
| Missing | 5 |
| Residence | |
| Rural | 75 (71.4) |
| Semi-urban | 30 (28.6) |
| Missing | 1 |
| Primary vaccine received | |
| Moderna | 57 (53.8) |
| Pfizer | 45 (42.5) |
| Janssen | 4 (3.8) |

**Table 2. Mean IgG and corresponding 95% confidence intervals at months 0, 3, and 6 by depressive symptom status estimated using GEE.**

| Month post-vaccination | Depressive symptoms (n = 14) | No depressive symptoms (n = 92) | Difference |
|---|---|---|---|
| Month 0 | 15.65 (6.49, 24.80) | 19.20 (12.33, 26.07) | −3.55 (−12.01, 4.90) |
| Month 3 | 17.00 (9.16, 24.85) | 18.00 (11.06, 24.94) | −1.00 (−7.33, 5.34) |
| Month 6 | 16.35 (6.46, 26.24) | 16.02 (8.46, 23.59) | 0.32 (−8.90, 9.55) |

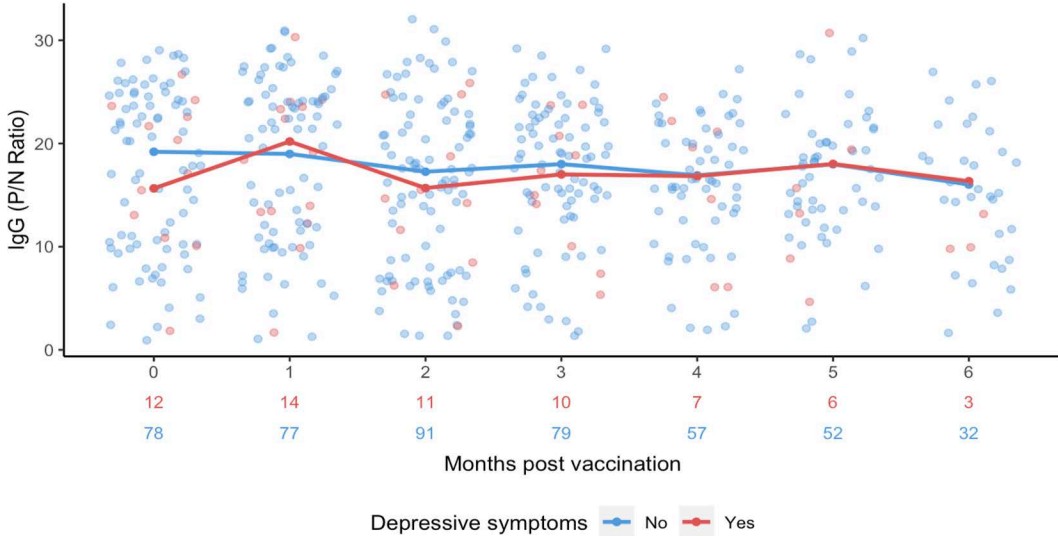

**Fig 2. Mean IgG estimated using GEE with multiple imputation from 0 to 6 months post-vaccination (solid lines) and non-imputed individual IgG samples (scatter plots) among individuals with depressive symptoms and individuals without depressive symptoms.** Sample size prior to imputation denoted for those with depressive symptoms (red) and those without (blue).

## 4. Discussion

We found that individuals with higher levels of depressive symptoms 4-week prior to their vaccination had lower mean IgG values immediately following vaccination than participants without depressive symptoms. However, the estimates of IgG were imprecise, possibly due to the small sample size and because IgG values in sera vary greatly from one individual to another. As such, the 95% confidence intervals for individuals with and without depressive symptoms overlap. After six months, IgG levels were similar between the two groups, although the number of samples provided were limited due to loss to follow-up. There is modest evidence that individuals with depressive symptoms have a smaller initial antibody response following vaccination compared to individuals without depressive symptoms, though the clinical relevance is unclear.

The COVID-19 pandemic has highlighted several health-related disparities in certain populations. People with mental health disorders, including depression, are one such group that experiences worse COVID-19 outcomes than others [13,22,23]. These individuals face an increased risk of COVID-19 infection and mortality. In this study, we found that depressive symptoms play a role in altering the initial response to COVID-19 vaccines. This aligns with the conclusions of previous studies that psychological stress and mental health disorders can alter initial vaccine response [35–37]. Individuals with depressive symptoms face multifactorial burdens related to COVID-19.

There are several strengths to the study. First, the study was conducted using a probability-based design, thus increasing the representativeness of our sample to Chatham County. Census blocks with higher concentrations of Hispanic/

Latino and/or Black/African American populations were intentionally over-sampled and lower-income communities and rural areas were included to increase the generalizability of the sample to Chatham County [28]. Second, the study measured both IgG in serum and depressive symptoms via survey at regular intervals during follow-up, and we leverage longitudinal IgG and depressive symptom data in our models. In addition, the frequency of surveys captures changes in mental health indicators, including depression.

The study does have limitations. Despite the probability-based design, our study sample may not fully represent the population of Chatham County, largely due to nonresponse bias. Door-to-door visits were conducted by study staff to households selected for inclusion in the study in efforts to increase the sample size, however, the study still enrolled fewer participants than expected. Nonresponse varied greatly by age, and as a result, the study sample consists mostly of older participants which impacts the generalizability of the sample. Additionally, the use of surveys does not allow for perfect measurement of depressive symptoms. For instance, participants may not answer sensitive questions truthfully. Lastly, there may be sources of unmeasured confounding of the relationship between depressive symptoms and antibody response that were not accounted for.

The use of the CESD-20 has its own set of limitations. First, this analysis only accounts for CESD-20 scores during a single period, in the four weeks prior to vaccination. Depression scores at other points in time may be associated differently with the outcome. Further, the performance of the CESD-20 screening tool may not have the same reliability in all populations [32,38]. In a meta-analysis of 28 studies, the CESD-20 with a cutoff ≥16 demonstrated higher pooled sensitivity than specificity (0.87 vs 0.70), highlighting that the tool will produce a greater number of false-positive than false-negative results [32]. Higher cutoffs have been suggested (e.g., ≥20), which have increased specificity at the expensive of decreased sensitivity [32]. Our study may be subject to both false-negative and false-positive classifications of depressive symptoms. Due to our small sample size, we were unable to evaluate other cutoffs, however, we reached similar conclusions with a dichotomized CESD-20 score and a continuous score.

The importance of antibody levels for protection against COVID-19 has yet to be established, so these results do not indicate whether depressive symptoms can alter the efficacy of the vaccines. Other research has suggested that those with mental health disorders may experience a decrease in COVID-19 vaccine effectiveness [12,13,18]. Another study, however, demonstrated that mRNA vaccines provide similar efficacy for individuals with and without psychiatric disorders [39], which may suggest that even a diminished immune response among individuals with depression may be sufficient for protection.

The results of this analysis provide evidence that further work is warranted to evaluate the association between mental health disorders and antibody response after COVID-19 vaccination. Future work should account for the longitudinal nature of both IgG in serum and CESD-20 scores. While this analysis focuses on depressive symptoms, the impact of other mental health disorders, such as anxiety and PTSD, should also be evaluated to determine their role in shaping antibody response after vaccination.

## Supporting information

**S1 Fig. Timeline detailing definition of months post-vaccination for two-dose mRNA vaccines (Moderna and Pfizer) and single-dose Janssen vaccine.** Created with BioRender.com.
(PNG)

**S1 Table. Parameter estimates for all covariates included in primary analysis model.** All covariates are binary.
(DOCX)

**S1 File. Baseline survey questions in English and Spanish for the Chatham County COVID-19 Cohort (C4) study.**
(PDF)

**S2 File. Biweekly survey questions in English and Spanish for the Chatham County COVID-19 Cohort (C4) study.**
(PDF)

## Author contributions

**Conceptualization:** Emily J. Ciccone, Ross M. Boyce, Bonnie E Shook-Sa, Allison E. Aiello.

**Data curation:** Annika K. Gunderson, Haley Abernathy, Haley E. Garrett, Elise King.

**Formal analysis:** Caitlin A. Cassidy.

**Funding acquisition:** Ross M. Boyce, Allison E. Aiello.

**Investigation:** Anne P. Starling, Emily J. Ciccone, Ross M. Boyce, Bonnie E Shook-Sa, Allison E. Aiello.

**Methodology:** Caitlin A. Cassidy, Jessie K. Edwards, Haley Abernathy, Haley E. Garrett, Elise King, Ross M. Boyce, Bonnie E Shook-Sa, Allison E. Aiello.

**Project administration:** Annika K. Gunderson, Cherese N. Beatty Parker, Anne P. Starling, Emily J. Ciccone, Ross M. Boyce, Bonnie E Shook-Sa, Allison E. Aiello.

**Supervision:** Cherese N. Beatty Parker, Anne P. Starling, Emily J. Ciccone, Ross M. Boyce, Bonnie E Shook-Sa, Allison E. Aiello.

**Visualization:** Caitlin A. Cassidy.

**Writing – original draft:** Caitlin A. Cassidy.

**Writing – review & editing:** Caitlin A. Cassidy, Jessie K. Edwards, Annika K. Gunderson, Haley Abernathy, Haley E. Garrett, Elise King, Cherese N. Beatty Parker, Anne P. Starling, Emily J. Ciccone, Ross M. Boyce, Bonnie E Shook-Sa, Allison E. Aiello.

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
