## [Decision Letter · Decision Letter 0]

23 Mar 2025

PMEN-D-25-00059

The association between depressive symptoms and antibody response following SARS-CoV-2 vaccination among central North Carolina residents

PLOS Mental Health

Dear Dr. Cassidy,

Thank you for submitting your manuscript to PLOS Mental Health. After careful consideration, we feel that it has merit but does not fully meet PLOS Mental Health’s publication criteria as it currently stands. Therefore, we invite you to submit a revised version of the manuscript that addresses the points raised during the review process.

We look forward to receiving your revised manuscript.

Kind regards,

Amin Tajerian, M.D.

Academic Editor

PLOS Mental Health

Journal Requirements:

1. We noticed you have some minor occurrence of overlapping text with the following previous publication(s), which needs to be addressed:

- https://doi.org/10.3389/fimmu.2022.965971 

- DOI: 10.1097/EDE.0000000000001625

- doi: 10.1136/bmj-2021-068302

In your revision ensure you cite all your sources (including your own works), and quote or rephrase any duplicated text outside the methods section. Further consideration is dependent on these concerns being addressed.

i. Please clarify all sources of funding (financial or material support) for your study. List the grants (with grant number) or organizations (with url) that supported your study, including funding received from your institution. 

ii. State the initials, alongside each funding source, of each author to receive each grant.

iii. State what role the funders took in the study. If the funders had no role in your study, please state: “The funders had no role in study design, data collection and analysis, decision to publish, or preparation of the manuscript.”

iv. If any authors received a salary from any of your funders, please state which authors and which funders.

3. Please make sure the funding information on the submission form matches your financial disclosure statement. Please indicate by return the full and correct funding information for your study and confirm the order in which funding contributions should appear. Please be sure to indicate whether the funders played any role in the study design, data collection and analysis, decision to publish, or preparation of the manuscript.

4. Please insert an Ethics Statement at the beginning of your Methods section, under a subheading 'Ethics Statement'.

5. Please provide separate figure files in .tif or .eps format.

https://journals.plos.org/mentalhealth/s/figures

https://journals.plos.org/mentalhealth/s/figures#loc-file-requirements

6. We have noticed that you have uploaded Supporting Information files, but you have not included a list of legends. Please add a full list of legends for your Supporting Information files after the references list.

7. In the online submission form, you indicated that “Deidentified individual data that supports the results will be shared beginning 9 to 36 months following publication upon request to the corresponding author, provided the investigator who proposes to use the data has approval from an Institutional Review Board (IRB), Independent Ethics Committee (IEC), or Research Ethics Board (REB), as applicable, and executes a data use/sharing agreement with UNC. Principal Investigator should submit a request to the UNC Industry Contracting team (OSPContracting@unc.edu) to initiate the data use/sharing agreement.”

3. Uploaded as supplementary information.

Additional Editor Comments (if provided):

The CESD-20 is designed to assess current depressive symptomatology experienced over the past two weeks in the general population. However, it does not capture the chronicity or functional impairment necessary for a clinical diagnosis of Major Depressive Disorder (MDD). While the CESD-20 demonstrates high sensitivity, making it effective in identifying individuals with depressive symptoms, it has only moderate specificity. This means it may occasionally classify individuals without clinically significant depression as positive, leading to potential false positives.

CESD-20 is useful for capturing short-term fluctuations in mood, making it well-suited for large-scale, cross-sectional studies. However, its limitations become more pronounced in longitudinal research or studies with small sample sizes, where the lack of information on symptom chronicity and functional impairment may reduce its utility and increase the risk of misclassification.

Please discuss how the CESD-20’s focus on current depressive symptoms is relevant to your study and address validity, reliability, sensitivity, and specificity of using CESD-20 >16 cutoff in COVID19 era.

The use of GEE with interaction terms may be too complex for the small subgroup.

Greater transparency in reporting statistical differences in IgG responses is recommended.

Consider modeling CESD-20 as a continuous variable and report the effects of covariates used to adjust the IgG response

Consider using a post hoc power analysis to objectively assess the impact of the small sample size on the study’s findings.

Kindly ensure that all reviewer comments are thoroughly addressed.

Reviewers' comments:

Reviewer's Responses to Questions

**Comments to the Author**

1. Does this manuscript meet PLOS Mental Health’s publication criteria? Is the manuscript technically sound, and do the data support the conclusions? The manuscript must describe methodologically and ethically rigorous research with conclusions that are appropriately drawn based on the data presented.

Reviewer #1: Partly

Reviewer #2: Partly

2. Has the statistical analysis been performed appropriately and rigorously?

Reviewer #1: Yes

Reviewer #2: Yes

3. Have the authors made all data underlying the findings in their manuscript fully available (please refer to the Data Availability Statement at the start of the manuscript PDF file)?

Reviewer #1: Yes

Reviewer #2: No

4. Is the manuscript presented in an intelligible fashion and written in standard English?

Reviewer #1: Yes

Reviewer #2: Yes

5. Review Comments to the Author

Reviewer #1: This study addresses an important and underexplored area of research, providing insights into how mental health factors may influence immune responses to vaccination. However, there are several areas where the study could be strengthened.

1-Statistical Power and Sample Size: Given the small number of participants with depressive symptoms (n=14), I suggest including a power calculation to assess whether the study is adequately powered to detect meaningful differences in antibody levels.

2-Measurement of Depressive Symptoms: The use of the CESD-20 is appropriate, but I suggest considering an analysis where depressive symptoms are treated as a continuous variable or stratified by severity, rather than a strict binary cutoff (≥16), to provide a more nuanced understanding of the association.

3-Interpretation of Findings: The overlapping confidence intervals in IgG levels between groups indicate that the observed differences might be due to variability and limited sample size. I suggest discussing whether these differences have clinical relevance or are primarily a statistical finding.

4-Missing Data and Imputation Strategy: Since multiple imputation was used for missing IgG values and covariates, I recommend providing more details about the assumptions made, how missingness was assessed, and whether sensitivity analyses were performed to ensure the robustness of the results.

5-Data Visualization: Figure 2 could be improved by adding clearer annotations and possibly overlaying individual data points on the model-predicted means to better illustrate trends.

6-Discussion on Public Health Implications: Expanding the discussion to consider potential interventions (e.g., psychological or behavioral strategies to improve vaccine efficacy in individuals with depressive symptoms) would strengthen the paper’s impact.

Reviewer #2: Thank you for submitting your interesting manuscript to this journal.

Methods:

Your study has a solid methodology, utilizing GEE and a reliable sampling method. You also explain the handling of missing data, which enhances the robustness of your work. My main concern is the disparity between your aim and methods compared to your results and discussion. I will address this in the following paragraphs.

Results:

Your demographic data is well described, although you did not include important baseline characteristics and comorbid conditions that could impact the immune response, such as diabetes mellitus, autoimmune diseases, immunosuppressant drug use, chemotherapy, and cancer. This bias is particularly critical in your study because your primary outcome is IgG, and your participants were predominantly older individuals.

You did not report any follow-up on depressive symptoms in your results, which contradicts the methods section, where you mention a biweekly survey to evaluate depressive symptoms. Additionally, you highlight this repetitive depressive symptom measurement in both your strengths and limitations sections, which is inconsistent.

The analysis report is unclear. You did not explain why you treated the CESD-20 score as a binary exposure. Moreover, you did not report the p-value or effect size for your model.

Discussion:

Attitudes and beliefs about vaccines are mentioned as a strength: “the frequency of surveys allows for monitoring of participant attitudes and beliefs towards vaccination”, yet there is no corresponding information about these surveys in the results or discussion sections.

The section in the discussion about COVID-19 outcomes seems irrelevant—why was it included?

Additionally, there is no conclusion section in your manuscript.

In conclusion, while this study is novel and interesting, the small sample size, methodological changes, and unclear analysis in the results section should be addressed before publication.

6. PLOS authors have the option to publish the peer review history of their article (what does this mean?). If published, this will include your full peer review and any attached files.

**Do you want your identity to be public for this peer review?** For information about this choice, including consent withdrawal, please see our Privacy Policy.

Reviewer #1: No

Reviewer #2: No

---

## [Decision Letter · Decision Letter 1]

21 Jul 2025

PMEN-D-25-00059R1

The association between depressive symptoms and antibody response following SARS-CoV-2 vaccination among central North Carolina residents

PLOS Mental Health

Dear Dr. Cassidy,

Thank you for submitting your manuscript to PLOS Mental Health. After careful consideration, we feel that it has merit but does not fully meet PLOS Mental Health’s publication criteria as it currently stands. Therefore, we invite you to submit a revised version of the manuscript that addresses the points raised during the review process.

We look forward to receiving your revised manuscript.

Kind regards,

Amin Tajerian, M.D.

Academic Editor

PLOS Mental Health

Journal Requirements:

Additional Editor Comments (if provided):

The statistical analysis, as presented, raises concerns regarding both integrity and transparency. Specifically, the claim of statistical significance at month 0 is not supported by the reported data. You state:

"Note, the 95% confidence interval for the difference does not include the null (Table 2, Fig 2)."

However, the confidence interval for the group difference at month 0 is -3.55 (95% CI: -12.01, 4.90), which does include the null hypothesis value of zero. This corresponds to an approximate p-value of 0.41, indicating no statistically significant difference between groups at that time point.

Moreover, all other confidence intervals for group differences (at months 3 and 6) also include zero, suggesting that these comparisons are likewise not statistically significant or meaningful.

It should not be the reader’s responsibility to reconstruct the statistical logic or determine significance from confidence intervals. It is the authors' responsibility to present results accurately, clearly, and without misrepresentation. As currently written, the reporting of statistical findings lacks the clarity and rigor.

I recommend revising the manuscript to reflect the correct interpretation of the data and ensure transparent communication of statistical findings.

Reviewers' comments:

Reviewer's Responses to Questions

**Comments to the Author**

1. If the authors have adequately addressed your comments raised in a previous round of review and you feel that this manuscript is now acceptable for publication, you may indicate that here to bypass the “Comments to the Author” section, enter your conflict of interest statement in the “Confidential to Editor” section, and submit your "Accept" recommendation.

Reviewer #1: All comments have been addressed

Reviewer #2: All comments have been addressed

2. Does this manuscript meet PLOS Mental Health’s publication criteria? Is the manuscript technically sound, and do the data support the conclusions? The manuscript must describe methodologically and ethically rigorous research with conclusions that are appropriately drawn based on the data presented.

Reviewer #1: Yes

Reviewer #2: Yes

3. Has the statistical analysis been performed appropriately and rigorously?

Reviewer #1: Yes

Reviewer #2: Yes

4. Have the authors made all data underlying the findings in their manuscript fully available (please refer to the Data Availability Statement at the start of the manuscript PDF file)?

Reviewer #1: Yes

Reviewer #2: Yes

5. Is the manuscript presented in an intelligible fashion and written in standard English?

Reviewer #1: Yes

Reviewer #2: Yes

6. Review Comments to the Author

Reviewer #1: (No Response)

Reviewer #2: (No Response)

7. PLOS authors have the option to publish the peer review history of their article (what does this mean?). If published, this will include your full peer review and any attached files.

**Do you want your identity to be public for this peer review?** For information about this choice, including consent withdrawal, please see our Privacy Policy.

Reviewer #1: **Yes: **Yalda Yazdani

Reviewer #2: No

---

## [Editor Report · Decision Letter 2]

5 Aug 2025

The association between depressive symptoms and antibody response following SARS-CoV-2 vaccination among central North Carolina residents

PMEN-D-25-00059R2

Dear Ms. Cassidy,

We are pleased to inform you that your manuscript 'The association between depressive symptoms and antibody response following SARS-CoV-2 vaccination among central North Carolina residents' has been provisionally accepted for publication in PLOS Mental Health.

Best regards,

Amin Tajerian, M.D.

Academic Editor

PLOS Mental Health

Thank you for your revision and addressing the concerns regarding the statistical interpretation.